

# Factors influencing healthcare provider respondent fatigue answering a globally administered in-app survey

Vikas N. O'Reilly-Shah

Department of Anesthesiology, Emory University, Atlanta, GA, United States of America
Department of Anesthesiology, Children's Healthcare of Atlanta, Atlanta, GA, United States of America

## ABSTRACT

**Background**. Respondent fatigue, also known as survey fatigue, is a common problem in the collection of survey data. Factors that are known to influence respondent fatigue include survey length, survey topic, question complexity, and open-ended question type. There is a great deal of interest in understanding the drivers of physician survey responsiveness due to the value of information received from these practitioners. With the recent explosion of mobile smartphone technology, it has been possible to obtain survey data from users of mobile applications (apps) on a question-by-question basis. The author obtained basic demographic survey data as well as survey data related to an anesthesiology-specific drug called sugammadex and leveraged nonresponse rates to examine factors that influenced respondent fatigue.

**Methods**. Primary data were collected between December 2015 and February 2017. Surveys and in-app analytics were collected from global users of a mobile anesthesia calculator app. Key independent variables were user country, healthcare provider role, rating of importance of the app to personal practice, length of time in practice, and frequency of app use. Key dependent variable was the metric of respondent fatigue.

**Results**. Provider role and World Bank country income level were predictive of the rate of respondent fatigue for this in-app survey. Importance of the app to the provider and length of time in practice were moderately associated with fatigue. Frequency of app use was not associated. This study focused on a survey with a topic closely related to the subject area of the app. Respondent fatigue rates will likely change dramatically if the topic does not align closely.

**Discussion**. Although apps may serve as powerful platforms for data collection, responses rates to in-app surveys may differ on the basis of important respondent characteristics. Studies should be carefully designed to mitigate fatigue as well as powered with the understanding of the respondent characteristics that may have higher rates of respondent fatigue.

Corresponding author
Vikas N. O'Reilly-Shah,
voreill@emory.edu

## INTRODUCTION

The explosion of smartphone technology (*Rivera & Van der Meulen*) that has accompanied the digital revolution brings opportunities for research into human behaviour at an

unprecedented scale. Mobile analytics (*Amazon, 2016*; *Google, 2016*; *Xamarin, 2016*; *Microsoft, 2016*), along with tools that can supplement these analytics with survey data (*Xiong et al., 2016*; *O'Reilly-Shah & Mackey, 2016*), have become easy to integrate into the millions of available apps in public app stores (*Statistic, 2015*). Overall growth in app availability and use has been accompanied by concomitant growth in the mobile health (mHealth) space (*Akter & Ray, 2010*; *Liu et al., 2011*; *Ozdalga, Ozdalga & Ahuja, 2012*). mHealth is an established MeSH entry term that broadly describes efforts in the area of mobile-based health information delivery, although a reasonable argument can be made that it includes the collection of analytics and metadata from consumers of this information as well (*HIMSS, 2012*; *National Library of Medicine, 2017*). Surveys are a critical supplement to these analytics and metadata because they provide direct information about user demographic characteristics as well as the opinion information that researchers are most interested in understanding. Much of the mHealth literature has made use of surveys deployed them via Web-based online surveys (e.g., via REDCap (*Harris et al., 2009*) or via SurveyMonkey (SurveyMonkey, Inc, San Mateo, CA, USA)) rather than in-app surveys. However, mobile applications that are used by specialist populations create an opportunity for collection of information from a targeted group via in-app surveys, and tools are being developed to assess the quality of these health mobile apps for even better targeting (*Stoyanov et al., 2015*).

Respondent fatigue, also known as survey fatigue, is a common problem in the collection of survey data (*Whelan, 2008*; *Ben-Nun, 2008*). It refers to the situation in which respondents give less thoughtful answers to questions in the later parts of a survey, or prematurely terminate participation (*Whelan, 2008*; *Ben-Nun, 2008*; *Hochheimer et al., 2016*). This may be detected when there is straight-line answering, where the respondent chooses e.g., the first option of a multiple choice survey for multiple question in a row. It may also be present if the respondent leaves text response fields blank or if the respondent chooses the "default" response on a slider bar. Finally, fatigue may be present if the respondent fails to complete the survey (*Ben-Nun, 2008*; *Hochheimer et al., 2016*). Factors that are known to influence respondent fatigue include survey length, survey topic, question complexity, and question type (open-ended questions tend to induce more fatigue) (*Ben-Nun, 2008*). Respondent fatigue lowers the quality of data collected for later questions in the survey and can introduce bias into studies, including nonresponse bias (*JSM, 2016*; *JSM, 2015*).

There is a great deal of interest in understanding the drivers of physician survey responsiveness due to the value of information received from these practitioners (*Kellerman, 2001*; *Cull et al., 2005*; *Nicholls et al., 2011*; *Glidewell et al., 2012*; *Cook et al., 2016*). These studies typically looked at overall response rate rather than respondent fatigue. The collection of survey data in mobile apps may be collected on a question-by-question basis (*O'Reilly-Shah & Mackey, 2016*). While this increases the amount of data available to researchers, it also increases the risk of obtaining incomplete survey data as it may become more commonplace for users to discontinue study participation in the middle of a survey.

While incomplete survey data reduces the quality of a dataset, it also provides an opportunity to study respondent fatigue directly. In the course of a study of more than
10,000 global users of a mobile anesthesia calculator app (*O'Reilly-Shah, Easton & Gillespie, in press*), the author obtained basic demographic survey data as well as survey data related to an anesthesiology-specific drug called sugammadex. Nonresponse rates were leveraged to examine factors that influenced respondent fatigue.

## METHODS

As described elsewhere (*O'Reilly-Shah, Easton & Gillespie, in press*), the author has deployed a mobile anesthesia calculator app fitted with the Survalytics platform (*O'Reilly-Shah & Mackey, 2016*). A screenshot of the app interface is provided in Fig. S1. The calculator is designed to provide age and weight based clinical decision support for anesthetic management, including information about airway equipment, emergency management, drug dosing, and nerve-blocks. Survalytics enables cloud-based delivery of survey questions and storage of both survey responses and app "analytics" using an Amazon (Seattle, WA, USA) Web Services database. Here, analytics is used to mean collected and derived metadata including app use frequency, in-app activity, device location and language, and time of use. Two surveys were deployed: one to collect basic user demographic information, and another to characterize attitudes and adverse event rates related to the drug sugammadex. These surveys are available for review in the Supplementary Data in Tables S1 and S2. Survey questions appear immediately after launch of the app, with a "Not Now/Answer Later" button, so if the user is needing to reference the app for emergency purposes then they can immediately go to the calculator without being forced to the answer the survey. Although data collection is ongoing, the study period for this work is limited to data collected between December 2015 and February 2017. The sugammadex survey was deployed in March 2016. The results of the sugammadex survey itself are beyond the scope of the present analysis, although preliminary results have been presented at a meeting and in correspondence (*O'Reilly-Shah, 2016*; *O'Reilly-Shah et al., in press*).

Raw data from the DynamoDB database were downloaded and processed using CRAN R (R Core Team, Vienna, Austria) v3.3 in the RStudio (RStudio Team, Boston, MA, USA) environment (*South, 2011*; *Arel-Bundock, 2014*; *Ooms, 2014*; *R Core Team, 2015*; *RStudio-Team, 2015*). User country was categorized using public World Bank classification of country income level (*World Bank, 2016*). In cases where users were active in more than one country, the country in which the most app uses were logged was taken as the primary country. Detailed information about the Survalytics package, the data collected for this study, and the approach to calculation of frequency of app use can be found in Appendix S1.

The study was reviewed and approved by the Emory University Institutional Review Board #IRB00082571. This review included a finding by the FDA that Anesthesiologist falls into the category of "enforcement discretion" as a medical device, meaning that, at present, the FDA does not intend to enforce requirements under the FD&C Act (*FDA & U.S. Department of Health and Human Services, Food and Drug Administration, Center for Devices and Radiological Health, Center for Biologics Evaluation and Research, 2015*).
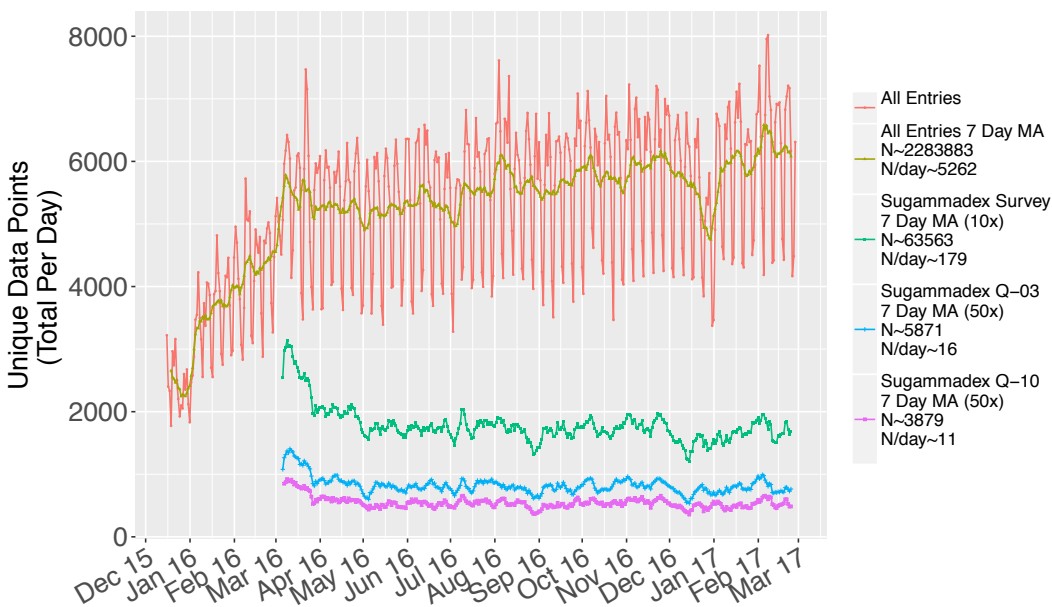

**Figure 1** Data collected over the study period, including visualization of the difference between rates of response to the first unbranched sugammadex survey (Q-03) and the final sugammadex survey (Q-10).

## Statistical methods

Subjects were categorized as "fatigued" or "not fatigued" according to whether they responded to the first unbranched sugammadex survey question (Table S2, Q-03) but did not complete the survey to the last question (Table S2, Q-10). This classification was used to perform logistic regression analysis against several independent variables, including provider role, frequency of app use, country income level, rating of app importance, and length of time in practice. Some of these were objectively gathered as metadata collected via the Survalytics package. Others were gathered from users as part of the basic demographic (Table S1).

## RESULTS

There was a consistent rate of data collection throughout the study period (Fig. 1). Following successful study launch in December 2015, the sugammadex survey was put into the field in March 2016. Responses to this survey were consistently collected throughout the study period, at a rate of approximately 179 total responses per day (green line, magnified 10×). There was a demonstrable and consistent rate of respondent fatigue, leading to the observed decrease in the rate of responses to the first unbranched question of the sugammadex survey (Q-03, blue line, magnified 50×, 16 responses collected per day) versus the last (Q-10, purple line, magnified 50×, 11 responses collected per day).

The overall rate of respondent fatigue was 34.3% ($N = 5,991$). Respondent fatigue then analyzed by several respondent characteristics. Some of these characteristics were based on self-reported data collected in the baseline survey (provider type, importance of app to

personal practice), while others were based on objective data (user location, frequency of app use). Results of univariable logistic regression analysis are shown in Table 1.

Provider role was an excellent predictor of the rate of respondent fatigue (Fig. 2, $N = 5{,}333$, $p < 0.001$). Physicians and physician trainees were most likely to complete the sugammadex survey, while technicians and respiratory therapists were least likely to do so. Main country income level was also an excellent predictor (Fig. 3, $N = 5{,}986$, $p < 0.001$); respondents from low income countries were less likely to complete the survey than those from high income countries.

Rating of the app's importance to the provider's practice was a moderate predictor of respondent fatigue (Fig. 4, $N = 3{,}642$, $p = 0.009$), although the relationship between app importance and respondent fatigue was unusual (see Discussion). Although length of time in practice had a statistically significant association with respondent fatigue (Fig. 5, $N = 2{,}518$, $p = 0.02$), the length of time in practice did not have monotonic directionality with regards to respondent fatigue. There was no association between the frequency of app use and respondent fatigue (Table 1, $N = 4{,}659$, $p = NS$).

## DISCUSSION

Overall, several provider characteristics, primarily provider role and World Bank country income level, were associated with the rate of respondent fatigue for an in-app survey. Other factors that would have been assumed to be associated with less respondent fatigue, such as higher frequency of app use, turned out not to be associated. Length of time in practice and and rating of importance of the app were associated with respondent fatigue, but the relationship was not linear with respect to the ordinal categorical responses. The initial part of the following discussion will focus on addressing the details of the findings, and the implications of each of these associations, in turn.

The association between provider role and fatigue rate is valuable because it demonstrates that researchers are likely to get a higher rate of complete response from users for whom the app and survey are well aligned. Physicians and anesthetists had the lowest rate of fatigue, and were users most likely to interact with the subject of the survey (sugammadex) on a frequent basis. Anesthesia techs and respiratory therapists are far less likely to use this drug or have knowledge of it, and so the high rate of observed respondent fatigue in these user groups is logical.

These findings extend previous work in the area of respondent fatigue in two ways. First, there do not appear to be any prior studies examining respondent fatigue for in-app mHealth surveys. Prior work examining respondent fatigue on the basis of unreturned surveys have found a highly variable rate of responsiveness, with response rates from physicians via Web-based methods as low as 45% (*Leece et al., 2004*) and as high as 78% (*Tran & Dilley, 2010*). Interestingly, there have been mixed findings as to the utility of Web-based methods over paper-and-pencil methods (*Leece et al., 2004*; *Nicholls et al., 2011*) although it is likely that the much older 2004 study from Leece et al. reflects a different era of connectedness. Given how much easier it is to quickly click through a survey on a mobile device as compared to filling out a pen-and-paper survey, or even sit
**Table 1 Univariable logistic regression results examining the association between various independent variables and the presence of respondent fatigue.** The association of respondent fatigue with provider role, country income level, rating of app importance, length of time in practice, and frequency of app use are discussed in greater detail in the text.

| | N (users) | N (fatigued) | Raw proportion of respondents with survey fatigue (%) | Odds ratio of being fatigued compared to referent category and 95% confidence interval | | | Proportion of respondents with survey fatigue (estimated percentage and 95% confidence interval) | | | Univariable *p*-value (overall wald per ind. var./vs reference category) |
|---|---|---|---|---|---|---|---|---|---|---|
| *Provider Role* | *5,333* | *1,708* | **32%** | | | | | *Overall Wald p-value =* | | **<0.001** |
| Physician | 1,832 | 467 | **25%** | **Referent** | Referent | Referent | **25%** | 24% | 28% | **Referent** |
| PhysicianTrainee | 1,331 | 343 | **26%** | **1.0** | 0.9 | 1.2 | **26%** | 23% | 28% | **0.85** |
| AA or CRNA | 1,488 | 553 | **37%** | **1.7** | 1.5 | 2.0 | **37%** | 35% | 40% | **<0.001** |
| AnesthesiaTechnician | 324 | 171 | **53%** | **3.3** | 2.6 | 4.2 | **53%** | 47% | 58% | **<0.001** |
| AA or CRNA Trainee | 210 | 88 | **42%** | **2.1** | 1.6 | 2.8 | **42%** | 35% | 49% | **<0.001** |
| Technically Trained in Anesthesia | 75 | 40 | **53%** | **3.3** | 2.1 | 5.3 | **53%** | 42% | 64% | **<0.001** |
| RespiratoryTherapist | 73 | 46 | **63%** | **5.0** | 3.1 | 8.2 | **63%** | 52% | 73% | **<0.001** |
| *Country Income* | *5,988* | *2,057* | **34%** | | | | | *Overall Wald p-value =* | | **<0.001** |
| Low income | 160 | 85 | **53%** | **Referent** | Referent | Referent | **53%** | 45% | 61% | **Referent** |
| Lower middle income | 1,172 | 548 | **47%** | **0.8** | 0.6 | 1.1 | **47%** | 44% | 50% | **0.13** |
| Upper middle income | 1,981 | 779 | **39%** | **0.6** | 0.4 | 0.8 | **39%** | 37% | 41% | **<0.001** |
| High income | 2,675 | 645 | **24%** | **0.3** | 0.2 | 0.4 | **24%** | 23% | 26% | **<0.001** |
| *Rating of App Importance* | *3642* | *956* | **26%** | | | | | *Overall Wald p-value =* | | **<0.001** |
| Absolutely Essential | 422 | 142 | **34%** | **Referent** | Referent | Referent | **34%** | 29% | 38% | **Referent** |
| Very Important | 1,174 | 306 | **26%** | **0.7** | 0.5 | 0.9 | **26%** | 24% | 29% | **0.003** |
| Of Average Importance | 1,123 | 230 | **20%** | **0.5** | 0.4 | 0.7 | **20%** | 18% | 23% | **<0.001** |
| Of Little Importance | 473 | 101 | **21%** | **0.5** | 0.4 | 0.7 | **21%** | 18% | 25% | **<0.001** |
| Not Important At All | 450 | 177 | **39%** | **1.3** | 1.0 | 1.7 | **39%** | 35% | 44% | **0.082** |
| *Length of Time in Practice* | *2518* | *685* | **27%** | | | | | *Overall Wald p-value =* | | **<0.001** |
| 0–5 Years | 1,030 | 285 | **28%** | **Referent** | Referent | Referent | **28%** | 25% | 30% | **Referent** |
| 6–10 Years | 489 | 99 | **20%** | **0.7** | 0.5 | 0.9 | **20%** | 17% | 24% | **0.002** |
| 11–20 Years | 448 | 109 | **24%** | **0.8** | 0.6 | 1.1 | **24%** | 21% | 28% | **0.18** |
| 21–30 Years | 551 | 192 | **35%** | **1.4** | 1.1 | 1.7 | **35%** | 31% | 39% | **0.003** |

O'Reilly-Shah (2017), *PeerJ*, DOI 10.7717/peerj.3785

**Table 1** (*continued*)

| | N (users) | N (fatigued) | Raw proportion of respondents with survey fatigue (%) | Odds ratio of being fatigued compared to referent category and 95% confidence interval | | | Proportion of respondents with survey fatigue (estimated percentage and 95% confidence interval) | | | Univariable *p*-value (overall wald per ind. var./vs reference category) |
|---|---|---|---|---|---|---|---|---|---|---|
| *Anesthesia Practice Model* | *2,951* | *709* | **24%** | | | | *Overall Wald p-value =* | | | **0.003** |
| Physician only | 1,040 | 249 | **24%** | **Referent** | Referent | Referent | **24%** | 21% | 27% | **Referent** |
| Physician supervised, anesthesiologist on site | 1,292 | 276 | **21%** | **0.9** | 0.7 | 1.0 | **21%** | 19% | 24% | **0.14** |
| Physician supervised, non-anesthesiologist physician on site | 189 | 55 | **29%** | **1.3** | 0.9 | 1.8 | **29%** | 23% | 36% | **0.13** |
| Physician supervised, no physician on site | 117 | 34 | **29%** | **1.3** | 0.8 | 2.0 | **29%** | 21% | 38% | **0.22** |
| No physician supervision | 170 | 46 | **27%** | **1.2** | 0.8 | 1.7 | **27%** | 21% | 34% | **0.38** |
| Not an anesthesia provider | 143 | 49 | **34%** | **1.7** | 1.1 | 2.4 | **34%** | 27% | 42% | **0.008** |
| *Practice Type* | *3,113* | *770* | **25%** | | | | *Overall Wald p-value =* | | | **<0.001** |
| Private clinic or office | 567 | 211 | **37%** | **Referent** | Referent | Referent | **37%** | 33% | 41% | **Referent** |
| Local health clinic | 277 | 88 | **32%** | **0.8** | 0.6 | 1.1 | **32%** | 26% | 37% | **0.12** |
| Ambulatory surgery center | 133 | 49 | **37%** | **1.0** | 0.7 | 1.5 | **37%** | 29% | 45% | **0.94** |
| Small community hospital | 330 | 65 | **20%** | **0.4** | 0.3 | 0.6 | **20%** | 16% | 24% | **<0.001** |
| Large community hospital | 932 | 183 | **20%** | **0.4** | 0.3 | 0.5 | **20%** | 17% | 22% | **<0.001** |
| Academic department/ University hospital | 874 | 174 | **20%** | **0.4** | 0.3 | 0.5 | **20%** | 17% | 23% | **<0.001** |

O'Reilly-Shah (2017), *PeerJ*, DOI 10.7717/peerj.3785

Peer J

**Table 1** (*continued*)

| | N (users) | N (fatigued) | Raw proportion of respondents with survey fatigue (%) | Odds ratio of being fatigued compared to referent category and 95% confidence interval | | | Proportion of respondents with survey fatigue (estimated percentage and 95% confidence interval) | | | Univariable *p*-value (overall wald per ind. var./vs reference category) |
|---|---|---|---|---|---|---|---|---|---|---|
| *Practice Size* | *3,342* | *880* | *26%* | | | | | *Overall Wald p-value =* | | *<0.001* |
| Solo | 1,364 | 426 | **31%** | **Referent** | Referent | Referent | **31%** | 29% | 34% | **Referent** |
| Small Group Less Than 10 | 695 | 223 | **32%** | **1.0** | 0.9 | 1.3 | **32%** | 29% | 36% | **0.69** |
| Medium Group 10–25 | 396 | 63 | **16%** | **0.4** | 0.3 | 0.6 | **16%** | 13% | 20% | **<0.001** |
| Large Group Greater Than 25 | 887 | 168 | **19%** | **0.5** | 0.4 | 0.6 | **19%** | 16% | 22% | **<0.001** |
| *Community Served* | *2,256* | *552* | *24%* | | | | | *Overall Wald p-value =* | | *<0.001* |
| Rural | 538 | 177 | **33%** | **Referent** | Referent | Referent | **33%** | 29% | 37% | **Referent** |
| Suburban | 374 | 91 | **24%** | **0.7** | 0.5 | 0.9 | **24%** | 20% | 29% | **0.01** |
| Urban | 1,344 | 284 | **21%** | **0.5** | 0.4 | 0.7 | **21%** | 19% | 23% | **<0.001** |
| *Frequency of App Use* | | | | | | | | *Overall Wald p-value =* | | *0.44* |
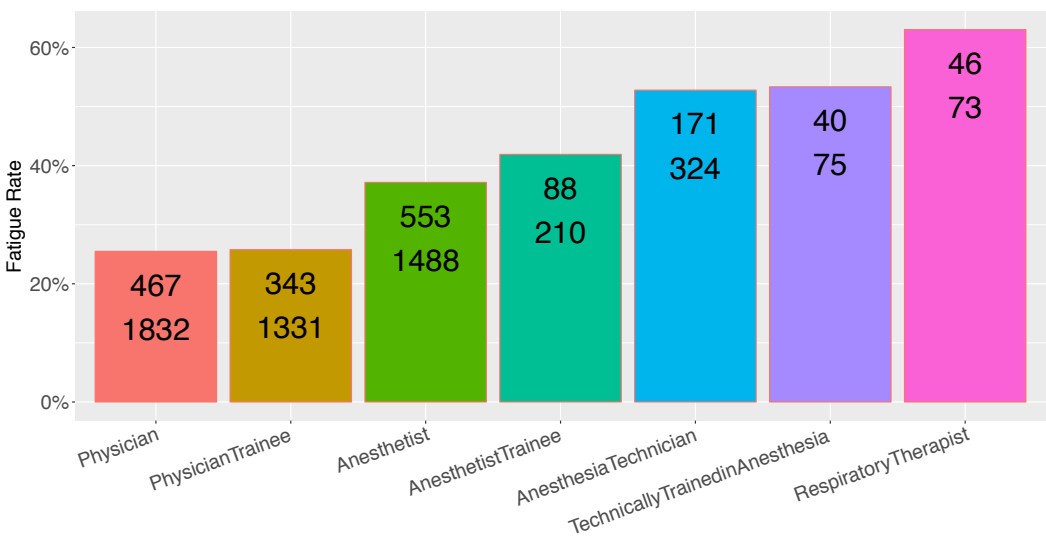

**Figure 2** **Observed fatigue rate versus provider role.** Top number is the number of participants with respondent fatigue (see 'Methods'). Bottom number is the total number of participants in the category.

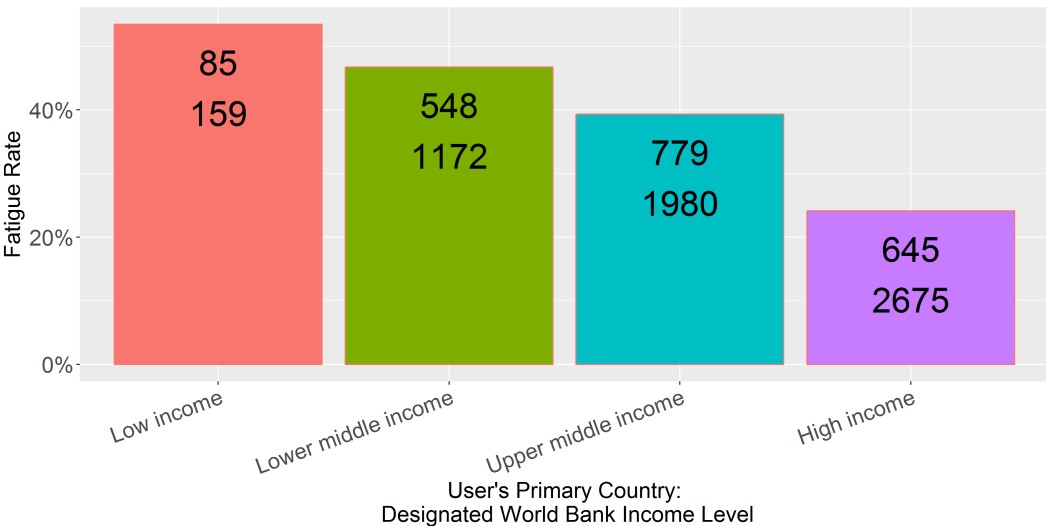

**Figure 3** **Observed fatigue rate versus primary country World Bank income level.** Top number is the number of participants with respondent fatigue (see 'Methods'). Bottom number is the total number of participants in the category.

down to a web survey provided via weblink, there would be no *a priori* reason to assume that respondent fatigue rates would be comparable. It is now estimated that two-thirds of time spent in the digital realm is time spent in mobile apps (*Bolton, 2016*). On the other hand, mobile apps are typically used in very short bursts, 2–8 min per session (*Average mobile app category session length 2015 | Statistic*). Apps, small programs with very specialized functions, are likely to be launched only when practically needed, potentially limiting the likelihood of participation in extraneous tasks such as in-app surveys.

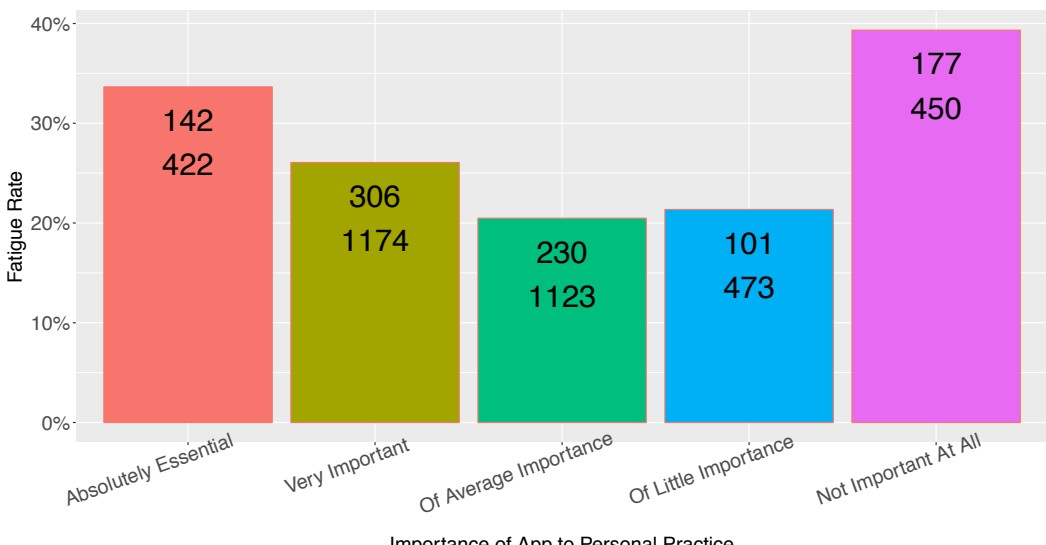

**Figure 4 Observed fatigue rate versus provider rating of app importance.** Top number is the number of participants with respondent fatigue (see 'Methods'). Bottom number is the total number of participants in the category.

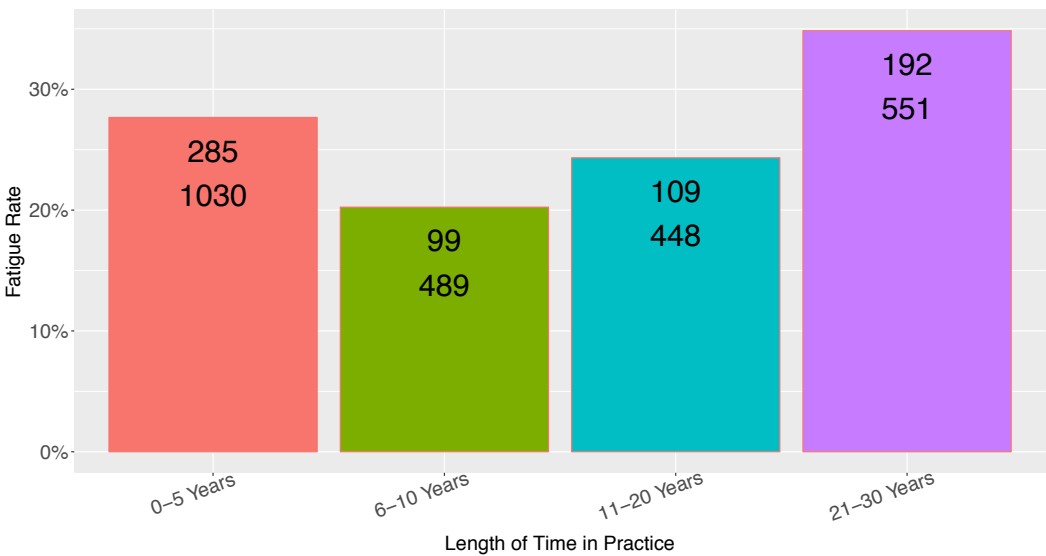

**Figure 5 Observed fatigue rate versus provider length of time in practice.** Top number is the number of participants with respondent fatigue (see 'Methods'). Bottom number is the total number of participants in the category.

Second, this study examines the rates of respondent fatigue during the course of a single survey, administered one question at a time, with full participant control over when to cease answering questions. Existing studies have primarily looked at global respondent fatigue in terms of e.g., rates of survey return. By allowing participants full control, the metadata revealed a more complete picture of the associations with respondent fatigue
**Table 2  Number and percentage of responses from users of each country income category within each category of responses regarding access to sugammadex.**

|  | Low income | | Lower middle income | | Upper middle income | | High income | |
| --- | --- | --- | --- | --- | --- | --- | --- | --- |
|  | N | % | N | % | N | % | N | % |
| Yes | 176 | 46% | 1,294 | 38% | 2,107 | 56% | 2,745 | 57% |
| No, not approved in my country | 84 | 22% | 760 | 22% | 544 | 14% | 436 | 9% |
| No, not on formulary | 57 | 15% | 775 | 23% | 613 | 16% | 956 | 20% |
| Yes, but not relevant to my practice | 23 | 6% | 222 | 6% | 250 | 7% | 233 | 5% |
| No or unsure, but not relevant to my practice | 40 | 11% | 366 | 11% | 257 | 7% | 406 | 9% |

during the course of a single survey, without needing to isolate phenomena such as straight-line answering.

The findings related to country income level are somewhat disheartening, in that the ability to reach and obtain feedback from users in resource-limited settings is a powerful promise of the global adoption of smartphones and mobile apps. Perhaps resource-limitations contributed the relatively high rate of respondent fatigue in users from lower-income countries: lack of access to reliable Internet connectivity (i.e., responses were recorded on the local device but not uploaded to the cloud), more expensive mobile data, and perhaps more time spent on patient care rather than in-app surveys. Another factor may be related to the expense of sugammadex itself; users from low-income countries were less likely to indicate access to sugammadex (Table 2, low = 46%, lower middle = 38%, upper middle = 56%, high income = 57%), and perhaps even users with access to it in lower-income countries did not feel they had enough experience with the drug to complete the survey.

The association of app importance to fatigue rate is interesting because it does not follow a monotonic trend, nor does it follow a pattern that would fit standard assumptions (Fig. 4). It is predictable that those viewing the app as "Not important at all" would have the highest rate of respondent fatigue, consistent with the present findings. Not intuitive, however, is the finding that users who rate the app as having average/little importance have the lowest rates of fatigue (meaning there was the highest rate of survey completion by those who rated the app as of middling importance to their practice). Perhaps those users who rate the app as more important to their practice take less time to complete the in-app survey because when they are using the app, they generally launch it for practical purposes. Likewise the association between length of practice and respondent fatigue does not follow a monotonic trend, which perhaps limits the usefulness of this finding in practice. It does suggest that the rate of responsiveness from providers early in their practice or with many years in practice may be reduced.

Premature termination of the survey was the approach used to measure survey fatigue for this study. This approach was chosen because this was an objective binary outcome that was straightforward to measure. For future studies, it may be possible to develop

alternative metrics of fatigue including a measurement of the rate of attrition at each question (were there step-offs?) or assessment of reduced thoughtfulness (was there straight-line answering?). This may lead to a conceptualization of survey fatigue on a spectrum rather than as a binary outcome. The attrition approach has been previously described for Web-based surveys (*Hochheimer et al., 2016*). More detailed analysis of step offs and straight-line answering may provide feedback for how to modify a survey to reduce the rate of fatigue. It isn't immediately clear how to objectively assess straight-line responses and more work will need to be performed to analyze and characterize these phenomena.

One limitation of these results is the lack of information about those respondents who chose to opt out of the study. Ethically, no demographic information about this population was possible. Those opting out of the study could be systematically biased in some way. Another limitation is that the survey topic was closely aligned with the subject area of the app. Respondent fatigue rates are likely to change dramatically if the topic does not align closely. This is supported in some ways by this data; as noted above, users who may have had less cause to use or interact with the drug were observed to have a much higher rate of respondent fatigue. The effect on response rates was dramatic; fatigue rates climbed to 60% for respiratory therapists who, one could speculate, would have less cause to interact with or administer sugammadex.

Overall, however, the population of users of this app are a self-selected group of providers with enough interest in anesthesiology management to download and use an app called "Anesthesiologist." Survey fatigue was measured only for those users who indicated that they had access to the drug and that it was relevant to their clinical practice (see Table S2; only users who answered "Yes" to Q-02 were presented the remainder of the survey, and fatigue was measured on the basis of answering Q-03 but not Q-10). Users who indicated it was not relevant to their practice ("Yes, but not relevant to my practice") were not presented Q-03 through Q-10 and were therefore *de facto* excluded from the present study group.

The approach to the analysis was to perform univariable regression on each factor rather than multiple regression. Multiple regression was avoided due to missingness in the data at a rate high enough that complete case regression of the dataset may yield biased responses, and also the rate of missingness may have yielded biased results following multiple imputation. Our concern is that if the missingness is not at random, then imputation would be an inappropriate approach as it would bias the sample in a similar manner to a complete case analysis. This missingness resulted from the approach to the demographic survey deployment, in which some questions were delayed in presentation to reduce the burden of the total survey load when initially opting into the study. This approach carried the benefit of reducing this burden but also introduced fatigue into the demographic survey itself. The presence of respondent fatigue in the demographic survey itself could result in non-random differences in populations exhibiting respondent fatigue at the sugammadex survey level. There is a hint that the missingness is nonrandom in that the raw respondent fatigue rates per category in the univariable analysis are different from one another—as high as 34% overall for country income level and as low as 24% for community served. If there is non-random missingness, then it becomes much more fraught to perform multiple

variable regression due to an inability to account for this confounder. This is a limitation of this data set that potentially also limits the generalizability of the findings.

## CONCLUSIONS

This study demonstrates some of the advantages and limitations of collected data from mobile apps, which can serve as powerful platforms for reaching a global set of users, studying practice patterns and usage habits. Studies should be carefully designed to mitigate fatigue as well as powered with the understanding of the respondent characteristics that may have higher rates of respondent fatigue. Variable rates of respondent fatigue across different categories of providers should be expected. The use of large-scale analytics will likely continue to grow, leading to crowdsourced sources of information. For example, researchers may use trend data from the from Google searches or from in-app clicks and surveys to detect outbreaks of disease. Other future studies may also survey providers about post-marketing drug-related adverse events. The ability to predict response rates, and therefore power these studies, will rely on an understanding of what factors may influence survey fatigue. The work presented here should help to elucidate some of that factors that influence respondent fatigue, as well as demonstrate the applicability of this methodology to measure these fatigue rates for in-app surveys for providers using mHealth apps.

## ACKNOWLEDGEMENTS

I'd like to thank George Easton, who developed the methodology for calculating frequency of app use but far more importantly has provided excellent conversation and discussion that has influenced my thinking leading to this work. I'd also like to thank Scott Gillespie, who provided the R code for mapping provider country of origin income level and helped me with my statistical thinking.

### Funding

The Emory University Department of Anesthesiology supported this work. The funders had no role in study design, data collection and analysis, decision to publish, or preparation of the manuscript.

### Grant Disclosures

The following grant information was disclosed by the author:
Emory University Department of Anesthesiology.

### Competing Interests

No support from any organization for the submitted work; no financial relationships with any organizations that might have an interest in the submitted work in the previous three years; no other relationships or activities that could appear to have influenced the submitted work. The app was initially released in 2011 by Vikas O'Reilly-Shah with advertising in the

free version and a paid companion app to remove the ads. The app intellectual property was transferred to Emory University in 2015 and advertisements were subsequently removed, and the companion app to remove ads made freely available for legacy users not updating to the ad-free version. Following review by the Emory University Research Conflict of Interest Committee, Vikas O'Reilly-Shah has been released from any conflict of interest management plan or oversight.

## Author Contributions

- Vikas N. O'Reilly-Shah conceived and designed the experiments, performed the experiments, analyzed the data, contributed reagents/materials/analysis tools, wrote the paper, prepared figures and/or tables, reviewed drafts of the paper.

## Human Ethics

The following information was supplied relating to ethical approvals (i.e., approving body and any reference numbers):

The study was reviewed and approved by the Emory University Institutional Review Board #IRB00082571.

## Data Availability

The raw data and code have been supplied as Supplementary Files.

## Supplemental Information

Supplemental information for this article can be found online at http://dx.doi.org/10.7717/peerj.3785#supplemental-information.

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
