# Peer review of "Factors influencing healthcare provider respondent fatigue answering a globally administered in-app survey"

_PeerJ, doi:10.7717/peerj.3785_

## Round 0.1 · original submission · Major Revisions

Based on 2 scientific experts in your field and my own review there was general enthusiasm for this well written manuscript. However, there was significant concern about the limited background provided and the limited information on the actual survey. Please consider these two overarching issues as you revise your manuscript. Other major and minor points can be found below and in each of the reviewers comments.

Additional Major Comments:

Why perform just univariable regression analysis? Why not perform a multivariable regression analysis to understand which of the factors is independently associated with response fatigue.

There is also huge variation on number who responded to each variable. Can that be explained in the text of the manuscript a bit?

An important issue with surveys is the generalizability given low response rates to even starting a survey. Is there any way to assess differences in who chooses to fill out any questions (opt-in) vs. who doesn’t fill out any questions?

There is useful information about working with app survey data, especially the GPS information. It seems important methodology information that might be of use to readers who want to use this methodology to evaluate their own apps.

The number of graphs could be reduced. Especially Figures 2-5, which will be redundant with Table 1 once it is reformatted.

Line 166 references crowd sourcing, but it too general to be meaningful. Please expand on this idea.

Additional Minor Comments:

Results section in the abstract has some incomplete sentence (for example last sentence ends in “,”.

Please include P-values for all results presented, even if not significant.

Binomial regression produces risk ratios not odds ratios. Please correct the language in the tables.

In Table 1: The raw regression coefficients are redundant with the RR presentation. I suggest putting the raw percentages of response fatigue as a first column and then the RR.

·

Basic reporting

1-The manuscript is written in clear and professional English language.
2-I can see mHealth as one of the listed keywords by the author, however, I would strongly suggest adding a brief description of mHealth use in conducting health-related surveys. Administering in-app survey is an interesting area, which should be elaborated in introduction section.
3-In first paragraph of the introduction, where definition of respondent fatigue is presented (Lines 40 and 41), please elaborate how “less thoughtful answers to the questions in later parts of a survey” can be assessed. Adding a line or two would be helpful.

Experimental design

Methods:
1-Though app features are described elsewhere, but it would still be useful to describe technology briefly for readers’ ease. Description and images of App interfaces may also be added.
2-In statistical methods section of the methods, researcher used termination of survey as approach to measure respondent fatigue. Is “less thoughtful answers” was used in assessing respondent fatigue? If no, then please explain why it was not measured.
3-It would be interesting to know how survey questions appear, while user is adhering to application workflow. What if, user misses any step of App workflow? Were there validation checks?

Validity of the findings

Results:
1-Result section lacks narrative overview or demographic details of the respondents. Please provide, if available
2-Line85: Methods section mentions Dec 2015 to February 2017 as data collection period. However, significance of March 2016 is unclear in the same regard.
3-Line 90 (correction): 11 per day >> 11 responses per day
Discussion and Conclusion:
1-The discussion is not positioned among the global literature. I would strongly suggest to cite existing work or literature around the subject.
2-In-survey provides an exciting avenue of mHealth discipline; however, it is important to discuss respondent fatigue in reference to other widely used surveying techniques or approaches, e.g. web-based survey, pen-and-paper survey. Support it with reported statistics.
3-To replicate the study design, I would suggest author to add a brief description on limitation of present study.
4-Please re-organize the content given in conclusion section. Most of the things mentioned in conclusion should belong to discussion section.

Additional comments

The paper provides quick and interesting read. Overall, it presents valuable aspect of in-app surveys with regards to respondent fatigue. However, I have few comments for author, so considering them would benefit the readers.

Reviewer 2 ·

Basic reporting

1. There seems to be a lack of literature referenced in regard to the background of app use in the field of mHealth. Situating your study in the field of mHealth will be important to your readers.

2. For example, Ozdalga, E., Ozdalga, A., & Ahuja, N. (2012). The smartphone in medicine: a review of current and potential use among physicians and students. Journal of medical Internet research, 14(5), e128.

3. And, Akter, S., & Ray, P. (2010). mHealth-an ultimate platform to serve the unserved. Yearb Med Inform, 2010, 94-100.

4. And, Stoyanov, S. R., Hides, L., Kavanagh, D. J., Zelenko, O., Tjondronegoro, D., & Mani, M. (2015). Mobile app rating scale: a new tool for assessing the quality of health mobile apps. JMIR mHealth and uHealth, 3(1), e27.

5. There is some visual noise in your bar graphs. I think the percentages to the right of the graphs is sufficient. Please see in-text comment.

Experimental design

1. Moving description of knowledge gap your manuscript addresses to the introduction will be beneficial to the reader, if not motivating.

2. I appreciated the methods section and the description of the independent variables.

3. Is there a spectrum to fatigue? In your definition there was only fatigued and not fatigued. This makes me wonder how many questions came after the first unbranched sugammadex questions and if you could develop more than a binary definition of the concept.

Validity of the findings

1. The manuscript is founded on a question that somewhat undermines the findings. The admission that some users of the app might not know, or need to know, about the drug sugammadex is problematic to the concept of fatigue. I think that you can only draw hard conclusions from groups that use the app that also need to know, and potentially use in some manner [administer, add to patient file], sugammadex.

2. The above point is critical in my mind, so focusing on physicians, physician trainees, anesthetist, anesthetist trainee, and high and upper middle income countries in the write up and in the tables will make your argument more convincing. I believe you will have power to do so.

Annotated reviews are not available for download in order to protect the identity of reviewers who chose to remain anonymous.

---

## Round 0.2 · Minor Revisions

An expert reviewer and myself, a member of the editorial board, have reviewed the revised manuscript. The expert reviewer had several minor comments that should be addressed including a clear definition of mHealth and information about why the sample included all users (in validity section).

The authors may also not be aware that the terminology binomial regression means different things in different disciplines. The response gave references for logistic regression (which would make you OR presentation correct). Why not just say you did logistic regression, which is widely understood across most disciplines in the health literature. In some health disciplines binomial regression implies a log link function (rather than a logit link) and corresponds to risk ratios rather than OR. Hence the confusion on the first review.

The response about missing data is slightly concerning if things really are missing not at random. Perhaps a few more sentences in the discussion are needed to clarify what types of biases the missing not at random induces.

Reviewer 2 ·

Basic reporting

1. There seems to be a lack of literature referenced in regard to the background of app use in the field of mHealth. Situating your study in the field of mHealth will be important to your readers. This was a comment on the first draft and I still believe it to be true here on this draft.

2. Regarding feedback from above, it seems the author cited several of the works that I suggested, but did not do a self-directed review of the examples I provided or the works cited in the examples. If this manuscript aims to inform the mHealth literature, more background needs to be provided.

3. If this paper does not intend to inform mHealth researchers by providing the background mentioned above, then an mHealth definition will be helpful to the anesthesiologists reading the paper.

4. Grammatically, there are many error with periods and citations, some coming before the citation, some coming after the citation.

5. In the discussion, does “missingness” mean missing data? Please clarify.

6. During discussion of fatigue, the phrase preliminarily terminate is used. Is this notion better communicated by the phrase prematurely terminate?

Experimental design

1. I disagree with the conclusion that "Not intuitive, however, is the finding that users who rate the app as having average/little importance have the lowest rates of fatigue.” Is it not true that a user that goes through the entire survey will have the best perspective on whether it is useful?

2. The description of why there is a binary, rather than a spectrum, conceptualization of fatigue is warranted and adds to the generalization of the argument.

Validity of the findings

1. The manuscript is founded on a question that somewhat undermines the findings. The admission that some users of the app might not know, or need to know, about the drug sugammadex is problematic to the concept of fatigue. I think that you can only draw hard conclusions from groups that use the app that also need to know, and potentially use in some manner [administer, add to patient file], sugammadex.

2. The above note was based on the first draft of the manuscript. The data set is large enough to only include end-users that need to know about sugammadex. Can you offer an explanation why all end-users are included in the data set if some do not need to know about the drug?

---

## Round 0.3 · accepted · Accept

No additional revisions are requested. The authors were responsive to the additional comments.